# EVOKE: EVOKING CRITICAL THINKING ABILITIES IN LLMS VIA REVIEWER-AUTHOR PROMPT EDITING

**Xinyu Hu**[♥][*] **Pengfei Tang**[♥] **Simiao Zuo**[♥] **Zihan Wang**[♦] **Bowen Song**[♣] **Qiang Lou**[♥]
**Jian Jiao**[♥] **Denis Charles**[♥]

[♥] Microsoft    [♦] University of Washington    [♣] University of Michigan

## ABSTRACT

Large language models (LLMs) have made impressive progress in natural language processing. These models rely on proper human instructions (or prompts) to generate suitable responses. However, the potential of LLMs are not fully harnessed by commonly-used prompting methods: many human-in-the-loop algorithms employ ad-hoc procedures for prompt selection; while auto prompt generation approaches are essentially searching all possible prompts randomly and inefficiently. We propose *Evoke*, an automatic prompt refinement framework. In *Evoke*, there are two instances of a same LLM: one as a reviewer (LLM-Reviewer), it scores the current prompt; the other as an author (LLM-Author), it edits the prompt by considering the edit history and the reviewer's feedback. Such an author-reviewer feedback loop ensures that the prompt is refined in each iteration. We further aggregate a data selection approach to *Evoke*, where only the hard samples are exposed to the LLM. The hard samples are more important because the LLM can develop deeper understanding of the tasks out of them, while the model may already know how to solve the easier cases. Experimental results show that Evoke significantly outperforms existing methods. For instance, in the challenging task of *logical fallacy detection*, *Evoke* scores above 80, while all other baseline methods struggle to reach 20.

## 1 INTRODUCTION

Consider an intriguing trio that at first glance seems unrelated: *bumble bees, cell phones, and exciting news*. At a superficial level, their commonality might note their plural forms; however, a more profound analysis reveals a shared essence: they all "create a buzz." This comparison sheds light on the depth and intricacy of human cognitive processes. At the heart of such processes is critical thinking, the ability to conceptualize, analyze, question, and evaluate ideas and beliefs. As we transition to the domain of artificial intelligence, it is observed that large language models (LLMs) have remarkably evolved as general problem solvers, urging us to ponder:

*Can LLMs think on their own?*

In practice, we observe that existing prompting methods are inadequate in evoking the critical thinking abilities of LLMs. For example, in Figure 1, we show two prompts for solving a common concept task. From Figure 1 (left), we see that for the input trio "*bumble bees, cell phones, and exciting news*", the LLM outputs a superficial common concept "*plural form*" using the hand-crafted prompt. On the other hand, with the prompt generated by the proposed method, the LLM demonstrates much deeper understanding about the task , i.e., it generates the correct answer "can cause a buzz" (see Figure 1, right). These results indicate that the quality of prompts are directly related to the performance of LLMs. In this work, we focus on prompting methods that enables LLMs to think on their own.

The current prompting methodologies exhibit significant drawbacks. Many prompting methods are ad hoc because of their human-in-the loop development paradigm. In such a process, given a target task, we first draft an initial prompt. Then, we refine the prompt using techniques such as chain-of-thought, few-shot demonstrations, and coding-style problem descriptions (Wei et al., 2022c;a;

---

[*]Corresponding author: xinyuhu@microsoft.com

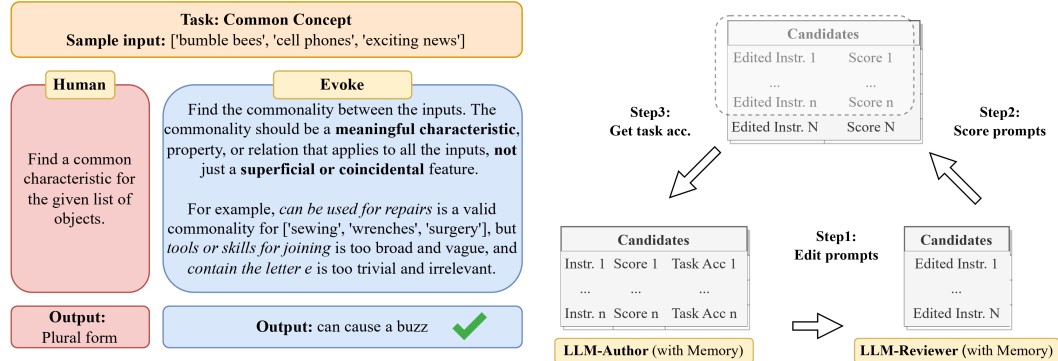

Figure 1: Comparison between hand-crafted and *Evoke* prompts.

Figure 2: Simplified workflow of *Evoke*.

Gao et al., 2023) based on the model's performance on the target task. We note that in practice, a hand-crafted prompt optimized for one task rarely translates to satisfactory performance in another task (Zhang et al., 2023). Therefore, each task becomes a new expedition, with its own set of trials, errors, and validations. Such an ad hoc human-in-the-loop development procedure introduces extensive human labor requirements, which significantly hinder the applicability of LLMs in real-world applications.

Existing works develop algorithms to automatically generate prompts instead of relying on ad hoc human optimization (Shin et al., 2020; Honovich et al., 2022; Zhou et al., 2022). However, these methods often lack feedback loops, such that the refinement procedure essentially performs a random search. For example, in each refinement iteration, Zhou et al. (2022) simply rephrases the prompt into multiple candidates, and then select the candidate that yields the best performance as the refined prompt. Note that such a procedure fails to learn from past successes and failures, such that refined prompt does not enrich the original prompt with additional context.

We propose *Evoke*, which addresses the aforementioned drawbacks by leveraging an author-reviewer paradigm. In this paradigm, there are two distinct purposes an LLM can serve: one instance as an author (LLM-Author) tasked with editing prompts, and another instance as a reviewer (LLM-Reviewer) tasked with evaluating the quality of the prompts generated by the LLM-Author. Each role is played independently by separate instances of the same LLM.

*Critical thinking is not something you do once with an issue and then drop it. It requires that we update our knowledge as new information comes in.*                                      **Daniel Levitin**

The essence of this quote resonates with the feedback loop in the workflow of *Evoke*, as depicted in Figure 2. The workflow comprises three steps: First, the LLM-Author edits prompts from previous iterations, taking into account the past edits and the feedback from the LLM-Reviewer. Second, the LLM-Reviewer scores the revised prompts from the LLM-Author, and the top-n candidates with the highest scores are selected for subsequent procedures. The LLM-Reviewer employs a memory module that stores history edits, prompts and task accuracy of history prompts. Finally, the task accuracy for each instruction is computed.

To further enhance the efficacy of *Evoke*, we propose a data selection strategy. In this strategy, only the *hard* samples selected by a selector are exposed to the LLM. The intuition is that the LLM can develop deeper understanding of the tasks out of the hard samples, while it already knows how to solve the easier cases. Through extensive experiments (see Figure 10 in the experiments), we see that retaining the hard samples indeed improves efficacy of *Evoke*.

We conduct extensive experiments to demonstrate the effectiveness of *Evoke*. Specifically, on eight tasks from the Instruction Induction (Honovich et al., 2022) dataset and the Big Bench Instruction Induction (Zhou et al., 2022) dataset, we show that *Evoke* significantly outperforms existing automatic prompt engineering approaches. For example, on the challenging logical fallacy detection task, *Evoke* achieves a score of over 80, while all the baseline methods struggle to reach 20. We also show that *Evoke* can improve LLMs' robustness against adversarial attacks, and can also handle fine-grained named entity recognition tasks with exceptional performance. As an example, *Evoke*

achieves significant performance gain on an adversarially constructed dataset, indicating that the proposed method can improve robustness of LLMs. Additionally, we provide detailed analysis on the effectiveness of each component of *Evoke*.

## 2 RELATED WORK

**Large Language Models** Recently, LLMs have shown emergent abilities—capabilities to perform tasks they weren't explicitly trained for (Wei et al., 2022a;b; Bubeck et al., 2023). This includes common sense question answering, code generation, and cross-domain problem solving, enriching their utility across unforeseen domains (Chen et al., 2021; Sarsa et al., 2022; Thirunavukarasu et al., 2023; Huang & Chang, 2022; Du et al., 2023). Subsequently, adapting LLMs to specific problems has drawn attention, and several methods have been proposed: Reinforcement Learning from Human Feedback (RLHF Ouyang et al. 2022), efficient fine-tuning (Hu et al., 2022; Dettmers et al., 2023), and prompt engineering (White et al., 2023), among others. Each method has its pros and cons. For instance, RLHF can significantly improve performance but may require extensive human annotations. Efficient fine-tuning, on the other hand, can be less resource-intensive but might fall short in achieving the desired level of task-specific optimization. Prompt engineering, while innovative, may require a well-crafted prompt to effectively guide the model towards accurate outputs.

**In-Context Learning and Prompt Engineering** In-Context Learning (ICL) refers to the ability of LLMs to learn a new task from a small set of examples presented within the context (the prompt) at inference time, without updating any parameters (Wei et al., 2022a). This paradigm has significantly improved the capabilities of LLMs across various tasks. Many studies have explored the reasons behind such improvements, examining aspects like Bayesian optimization and the difficulty of demonstrations (Xie et al., 2022; Min et al., 2022; Liu et al., 2022; Yoo et al., 2022).

Prompt engineering plays a pivotal role in facilitating ICL. It entails the design of prompts that arm the LLM with the essential information needed to learn and adeptly perform the new task. Each prompt essentially sets the stage for the LLM, enclosing the task's requirements and guiding the model towards producing the desired output. By carefully crafting prompts, it is possible to leverage the inherent capability of LLMs, enabling them to tackle a wide range of tasks even with limited or no prior explicit training on those tasks. Recently, methods such as Chain-of-Thought (CoT), Zero-CoT, Self-Consistency, Program-Aided, and Few-Shot Prompting have been demonstrated to be effective (Wei et al., 2022c; Kojima et al., 2022; Wang et al., 2022; Gao et al., 2023; Reynolds & McDonell, 2021).

**Automatic Prompt Engineering** The existing methodologies for automating discrete prompt optimization have their roots in instruction induction, as discussed by Honovich et al. 2022. It was discovered that LLMs can generate natural language instructions based on a small number of input-output pair examples. Building on this, Zhou et al. (2022) proposed a new algorithm for the automatic generation and selection of instructions for LLMs. The algorithm, named Automatic Prompt Engineer (APE), is capable of generating prompts that achieve human-level performance across a diverse range of NLP tasks. Work has also been done on automating prompt generation for specific domains like code generation, as discussed in Shrivastava et al. 2023.

## 3 ITERATIVE REVIEWER-AUTHOR PROMPT EDITING

### 3.1 OVERVIEW

In *Evoke*, the same LLM plays two different roles: an author (LLM-Author) that is in charge of editing and refine prompts, and a reviewer (LLM-Reviewer) that is in charge of scoring the refined prompts. We use two different prompts for the author's and the reviewer's task.

◇ **LLM-Author edits and generates new prompts based on feedback from LLM-Reviewer.** The prompt for LLM-Author consists of several components:

a **Input for editing:** Current task instruction to be refined and training data;

b **Instruction for editing:** "*We've provided pairs consisting of inputs, the teacher's correct answers, and the students' responses. Please review the incorrect responses from the stu-*

---

**Algorithm 1:** *Evoke*

---

**Require:** Training set; Initial prompt for the target task (i.e., the one we want to refine).
`// Initialization`
LLM-Selector: Initialize data scoring instruction.
LLM-Author: Initialize prompt editing instruction.
LLM-Review: Initialize prompt reviewing instruction.
**while** $t \leq T$ **do**
 |  `// LLM-Selector`
 |  Assign difficulty scores for each data point in the training set.
 |  Select a training subset based on the difficulty level.
 |  `// LLM-Author`
 |  LLM-Author generates multiple prompts based on the training data and its own memory.
 |  `// LLM-Reviewer`
 |  LLM-Reviewer scores the quality of each generated prompt from LLM-Author based on its own
 |   memory.
 |  Select top-n prompts based on the generated scores from LLM-Reviewer.
 |  Get task accuracy for all prompts.
 |  `// Memory update`
 |  Memory of LLM-Author appends (*edits*, *scores*).
 |  Memory of LLM-Reviewer appends (*edits*, *prompts*, *task accuracy*).
**Return:** The prompt with the highest task accuracy.

---

   *dents and summarize key points that could be adjusted in the instruction to enhance student accuracy. Highlight major edits and present the updated task instruction.*";

 c **Memory:** prior history *(edits, scores)*.

LLM-Author refines the instructions (prompts for the given task) by utilizing the training data and a memory component. We note that the memory consists of all prior *(edit, score)* pairs, where the *score* comes from LLM-Reviewer. This memory component enables LLM-Author to execute increasingly effective edits, drawing upon feedback from previous edits.

⋄ **LLM-Reviewer scores the quality of prompts generated by LLM-Author.** The input prompt for LLM-Reviewer consists of several components:

 a **Input for scoring:** problem description and current instruction from LLM-Author;

 b **Instruction for scoring:** "*Please rate the following instruction on a scale of 1 to 10, where 10 represents the highest level of clarity in problem description, execution steps, and a comprehensive explanation of the problem.*";

 c **Memory:** prior *(edits, instructions, task accuracy)*.

The instructions generated by LLM-Author are forwarded to LLM-Reviewer for evaluation. Based on the scores generated by LLM-Reviewer, only a subset of high-scoring candidates is selected to move on to the subsequent iteration. Through this iterative editing process between LLM-Author and LLM-Reviewer, LLM-Author can refine instructions in each iteration. Details of the algorithm can be found in Algorithm 1.

To illustrate the effectiveness of *Evoke*, first three edits from the Movie Recommendation task in Big Bench are presented in Figure 3. To start with, the prompt contains the basic task instruction. Next, it extracts key factors considered in movie recommendation, such as the genre of each movie, the distance between the given movies and the movies the user has watched before, and the popularity of the movies. In the final step, a well-explained example is presented with a detailed explanation following aforementioned factors. In summary, *Evoke* successfully concludes the key components of movie recommendation, and curates a demonstration with detailed explanation.

## 3.2 DATA SELECTION VIA LLM-SELECTOR

In practice, we find that not all samples are equally important to model performance (see Figure 10). In particular, we find that even without prompt refinement, the LLM already knows how to solve

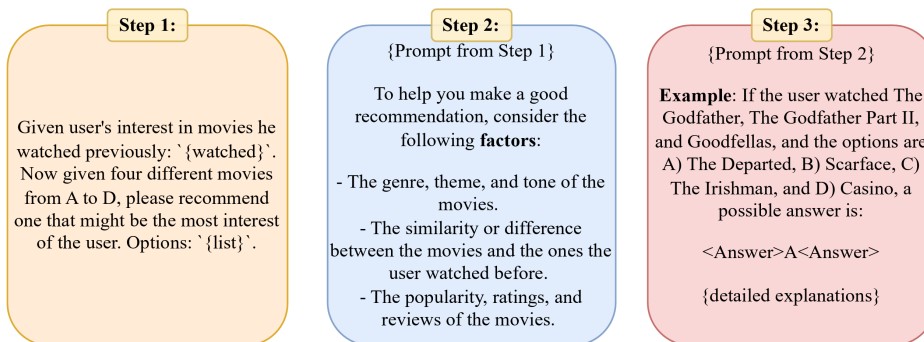

**Step 1:**

Given user's interest in movies he watched previously: `{watched}`. Now given four different movies from A to D, please recommend one that might be the most interest of the user. Options: `{list}`.

**Step 2:**

{Prompt from Step 1}

To help you make a good recommendation, consider the following **factors**:

- The genre, theme, and tone of the movies.
- The similarity or difference between the movies and the ones the user watched before.
- The popularity, ratings, and reviews of the movies.

**Step 3:**

{Prompt from Step 2}

**Example**: If the user watched The Godfather, The Godfather Part II, and Goodfellas, and the options are A) The Departed, B) Scarface, C) The Irishman, and D) Casino, a possible answer is:

<Answer>A<Answer>

{detailed explanations}

Figure 3: Illustration of Prompt Editing for the first three steps in the Task of Movie Recommendation within Big Bench.

some "easier" cases. Therefore, we only use "hard" samples in each refinement iteration. Specifically, we assign a third role besides an author a reviewer to the LLM: a data selector. The LLM-Selector evaluates the difficulty level (on a scale of 1 to 10) of each data point by assessing, based on the current task instruction, how challenging it is to derive the correct answer from the input. The input prompt for LLM-Selector consists of several components:

a **Input for evaluating difficulty level:** current instruction and input-output pair;

b **Instruction for evaluating difficulty level:** "*As an experienced teacher with insight into the various levels of difficulty of exam questions, please rate the following question on a scale of 1 to 10, considering factors such as conceptual understanding, application of knowledge, problem-solving skills, time required, clarity of language, and accessibility, where 1 denotes extremely easy and 10 denotes extremely difficult.*".

Empirically, we can further improve effectiveness of *Evoke* by using such a data selection strategy.

## 4 EXPERIMENTS

We conduct extensive experiments to demonstrate the effectiveness of *Evoke*. We show that for any given task, the prompts generated by *Evoke* include clear definitions and well-structured task execution steps. Moreover, these prompts feature demonstrations accompanied by detailed explanations. In all experiments, we utilize the Azure OpenAI API service (GPT-4) for the involved LLMs.

### 4.1 MAIN RESULTS

**Datasets.** We perform a comprehensive evaluation on eight tasks from Instruction Induction (Honovich et al., 2022) and Big Bench Instruction Induction (BBII) (Zhou et al., 2022), including

*orthography starts with*: Extract the words starting with a given letter from the input sentence.

*common concept*: Find a common characteristic for the given objects.

*rhymes*: Write a word that rhymes with the input word.

*movie recommendation*: Recommend movies similar to the given list of movies.

*logical fallacy detection*: Detect informal and formal logical fallacies.

*presuppositions as nli*: Determine whether the first sentence entails or contradicts the second.

*winowhy*: Evaluate the reasoning in answering Winograd Schema Challenge questions.

*epistemic reasoning*: Determine whether one sentence entails the next.

These tasks covers a wide range of natural language understanding, reasoning and inference tasks. For each task, we divide the dataset randomly into two sets, 60% of the data is allocated for training (prompt refinement) and the remaining 40% is for testing (prompt evaluation).

**Baselines.** We compare our methods against two baselines: human curated prompts (Human) from Honovich et al. (2022); Suzgun et al. (2022) and automatic prompt engineer (APE) proposed in (Zhou et al., 2022). APE first deduces an initial prompt from input-output pairs, and subsequently employs LLMs to refine and generate new prompt candidates. However, prompts are simply paraphrased during the refinement process of APE, which largely resembles random searching in the space of all possible prompts.

**Main Results.** Figure 4 demonstrates experimental results. We observe that *Evoke* outperforms all the baselines in all eight tasks. For example, on the challenging logical fallacy detection task from BBII, performance of *Evoke* is more than 80, while performance of both APE and Human are below 20. This is because *Evoke* is adept at conceptualizing the core definition of a task, decomposing a complex task into smaller subtasks, and curating relevant demonstrations accompanied by detailed explanations. To demonstrate the power of *Evoke*, we show the generated prompt for logical fallacy detection in Table 1. We see that the prompt begins with a clear task introduction and objective, followed by a fine-grained definition of logical fallacy. It then articulates the criteria for evaluation and the task steps to follow. Lastly, it provides a list of common logical fallacies, each accompanied by a detailed description. Additionally, a well-structured prompt for epistemic reasoning is presented in Table 2.

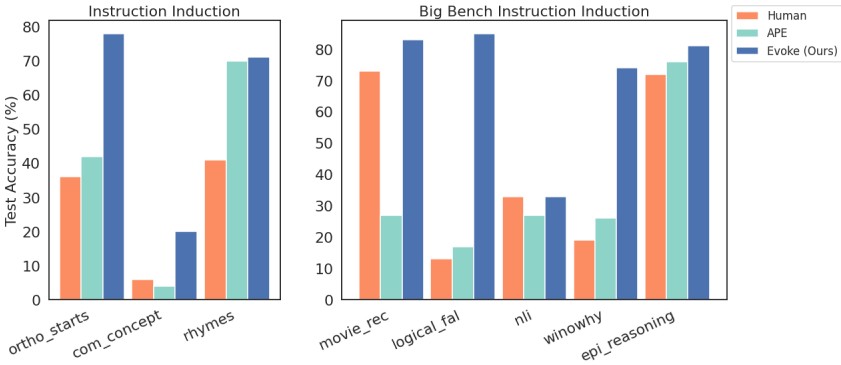

Figure 4: Results on eight tasks from the Instruction Induction and the Big Bench Instruction Induction datasets. We report the average accuracy over three runs.

## 4.2 Towards Adversarial Robustness

Despite their superior performance, LLMs are not robust to adversarial attacks (Wang et al., 2023). For example, when asking GPT-4 whether "pretty" is a positive word, the model can output the correct answer. However, if we ask whether "prettye", a clear typo of "pretty", is a positive word, the LLM outputs an opposite answer. We show that *Evoke* can generate prompts which alert the LLM in paying attention to potential typos, and thus can improve model robustness.

**Datasets.** We adopt two datasets: SST-2 (Socher et al., 2013) is a sentiment classification task, where we need to decide whether a movie review is positive or negative; and QQP (Wang et al., 2019) is a task where we need to determine whether two sentences are paraphrases of each other.

To evaluate whether *Evoke* can improve LLMs' robustness, we add typos to the datasets. Specifically, we perform character-level adversarial attacks for each sam-

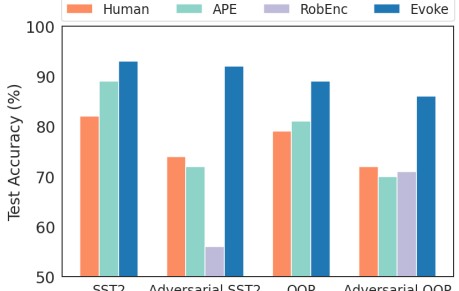

Figure 5: Results on clean and adversarially attacked SST2 and QQP datasets. We report the average accuracy over three runs. We note that RobEnc is only applied to the attacked data.

ple. In the attack, we change at most one character in each word, and we change at most 4 words in each sentence (Jones et al., 2020). In this way, the constructed adversarial texts are human-

Table 1: Prompt generated by *Evoke* for logical fallacy detection.

| Prompt from Evoke | Breakdown |
|---|---|
| In this task, you are required to identify both informal and formal logical fallacies in the provided input statements. Your response should be a binary value: return 1 if the query is logically valid (i.e., free from any logical fallacies), and return 0 if the query is logically invalid (i.e., contains at least one logical fallacy). | Introduction and objective |
| A logical fallacy refers to an error in reasoning. Informal fallacies are often content-dependent, such as appealing to irrelevant authority or making hasty generalizations. Formal fallacies, on the other hand, are structural errors in reasoning that occur regardless of the content. | Definition of logical fallacy |
| It is crucial to consider the structure and the substance of the argument, evaluating whether the conclusions follow logically from the premises, and whether the premises and assumptions are factual and valid. Be cautious not to let personal beliefs interfere with your analysis. | Evaluation criteria |
| For each given pair, compare the input statement against the principles of logical reasoning, to determine whether it contains a logical fallacy or not. Ensure your answer reflects the presence or absence of logical fallacies, thus determining the logical validity or invalidity of the statement. | Task steps |
| Here are some common examples of logical fallacies:
- Ad Hominem: {details}
- Appeal to Nature: {details}
- Hasty Generalization: {details}
- Post Hoc: {details}
- False Cause: {details} | Common examples of logical fallacy |

Table 2: Prompt generated by *Evoke* for epistemic reasoning.

| Prompt from Evoke | Breakdown |
|---|---|
| In this task, your goal is to determine whether the statement in the "Hypothesis" logically follows from the statement in the "Premise." This is known as entailment. If the "Hypothesis" statement is a logical consequence of the "Premise" statement, then it is an entailment. If it is not, then it is a non-entailment. | Introduction and objective |
| -Make sure to carefully consider the relations and assumptions mentioned in both the "Premise" and the "Hypothesis" statements.
-The entailment does not depend on the truth of the statements, but rather whether the logic in the "Hypothesis" follows from the "Premise".
-Pay close attention to the wording and structure of the sentences to analyze whether one entails the other. | Guidelines |
| Examples:
*Entailment*
Premise: The sun rises in the east.
Hypothesis: The sun rises.
Explanation: The Hypothesis is a simplified version of the Premise and does not introduce any new information or contradictions, hence it's an entailment.
*Non-entailment*
Premise: Sarah believes that all cats are black.
Hypothesis: All cats are black. Explanation: Even though the Hypothesis is expressed in the Premise, it's tied to Sarah's belief and not presented as a fact, hence it's a non-entailment. | Examples |
| Now, review the provided pairs of statements. Determine if the Hypothesis logically follows from the Premise and respond with either entailment or non-entailment. | Task Execution |

interpretable and simulate real typos. As an example, one sample from SST-2 is "*that's pure pr hype*", and its corresponding adversarial (corrupted) sample after the attack is "*tha'cs pure pr hyp*". We evaluate performance of different prompting methods on the corrupted samples.

**Baselines.** Besides APE and *Evoke*, we evaluate another model: RobEnc (Jones et al., 2020), which is a widely-used rule-based defense approach. RobEnc works as a clustering denoiser to cluster and

**APE**

Provided with a user query, please conduct a sentiment analysis. In this analysis, a score of 0 represents negative sentiment, while a score of 1 represents positive sentiment. Ensure to carefully check for any typos before providing a response.

**Evoke**

Now given a user query, please do a sentiment analysis where 0 represents negative sentiment and 1 represents positive sentiment. Please note that the input may contain typos. These are not intentional and do not reflect the user's actual sentiment. You should try to **correct the typos or infer the intended meaning from the context** before assigning a sentiment score. For example, `to merely badv trather than painfzully awfl` **could be corrected** to `to merely bad rather than painfully awful` and scored as 0. Similarly, `l comelling mobion` could be corrected to `a compelling motion` and scored as 1.

Figure 6: Prompts from APE and *Evoke* on adversarial attacked SST-2 task

denoise potentially corrupted inputs into an encoding, and then the denoised encoding is fed to the subsequent model (e.g., GPT-4) for inference. RobEnc learns rule-based word cluster for denoising: for example, if the word "hallo" is clustered around the word "hello", then all the "hallo" in the input will be converted to "hello".

**Results.** Figure 5 summarizes experimental results. We observe that *Evoke* significantly outperforms all the baselines in all the tasks. The performance gain is more significant for adversarially constructed datasets, e.g., Adversarial-SST2 and Adversarial-QQP. To understand this, we show the prompts generated by APE and *Evoke* in Figure 6. We see that although the prompt from APE provides a clear instruction regarding the given task and acknowledges the existence of typos, it does not provide clear guidelines on how to address the typo. On the other hand, the prompt from *Evoke* provides detailed explanations and actionable suggestions about defending against typos.

### 4.3 Towards Fine-Grained Tasks: Named Entity Recognition

Tasks in Figure 4 and Figure 5 are all sentence-level classification tasks, e.g., deciding whether a sentence is of positive or negative sentiment. In this section, we investigate whether *Evoke* can handle more fine-grained tasks, such as token-level named entity recognition (Schneider et al., 2020; Zuo et al., 2023).

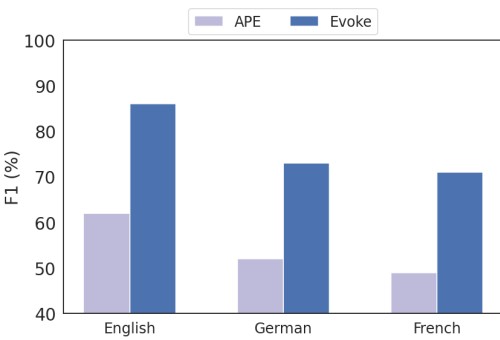

Figure 7: Results of APE and *Evoke* on an in-house multi-lingual NER dataset.

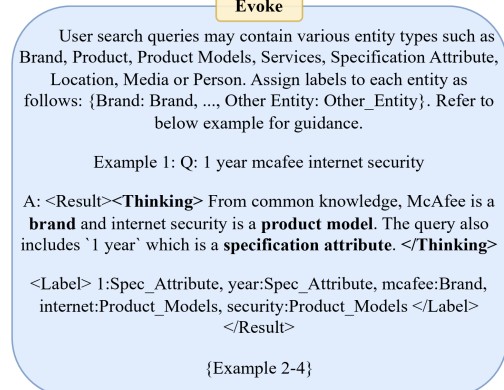

Figure 8: Prompt from *Evoke* on the NER task.

We collect multi-lingual in-house query data from a search engine, and for each token in the query, our goal is to assign the token to a pre-defined class (e.g., brand, location).

We illustrate results of *Evoke* on the fine-grained NER task in Figure 7. We see that *Evoke* significantly outperforms APE on all the languages. We further show the prompt generated by *Evoke* in Figure 8. From the prompt, we see that *Evoke* is able to automatically generate examples and explanations about the task.

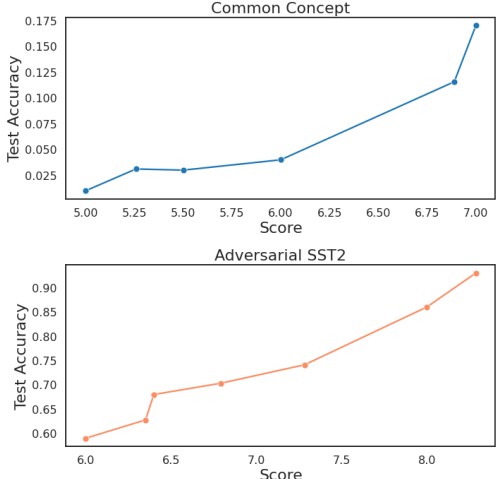

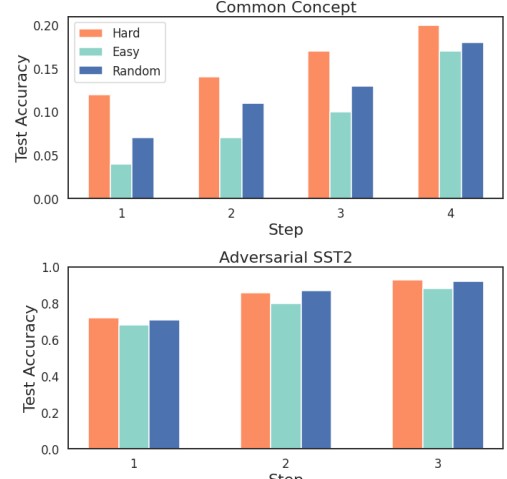

Figure 9: Correlation between scores generated by LLM-Reviewer and task accuracy.

Figure 10: Task accuracy over the number of iteration steps.

## 4.4 ANALYSIS

**LLM-Reviewer can judge the quality of prompts.** Recall that in *Evoke*, LLM-Reviewer scores all the prompts generated by LLM-Author. We empirically show that the scores can reflect the quality of the generated prompts. To examine the effectiveness of these scores, we illustrate the relationship between the scores and the task accuracy in Figure 9. In the experiments, we consider two tasks: Adversarial-SST2 and Common-Concept. From the results, we see that the scores can indeed reflect the final task accuracy. For example, for Common-Concept, we see that the task accuracy is about 5% when the prompt score is 6, and the task accuracy increases to about 17% when the prompt score increases to 7. A similar trend is also revealed on the Adversarial-SST2 task. We see that when the score is 7.5, the final task accuracy barely reaches 75%. And when the score increases to 8, the task accuracy increases to 85%.

**LLM-Author iteratively improves prompt generation.** In *Evoke*, because LLM-Author takes the feedback from LLM-Reviewer into consideration, it can iteratively improve the generated prompt. We demonstrate this in Figure 10 (the left-most orange bars). From the results, we see that indeed the final task accuracy continues to increases when we increase the number of iteration steps. For example, on Adversarial-SST2, with one refinement iteration, the final task accuracy is about 75%. When we increase the number of refinement iterations to 3, we see that task accuracy significantly increases to above 90%.

**Effectiveness of LLM-Selector.** Recall that in *Evoke*, we only consider the "hard" samples in each iteration. We demonstrate the effectiveness of such a strategy in Figure 10. We consider three settings: *Hard* is the strategy that we adopt in *Evoke*; *Random* is when we randomly select samples instead of selecting based on a score; and *Easy* is when we select the easy samples instead of the hard ones. From the results, we see that on both Common-Concept and Adversarial SST-2, *Easy* yields the worst performance, indicating that the hard samples are more helpful than the easy ones. Moreover, we observe that performance of *Random* is worse than *Hard* (i.e., *Evoke*), further implying the effectiveness of the proposed data selection strategy.

## 5 CONCLUSION

We propose *Evoke*, an author-reviewer framework for automatic prompt engineering. In *Evoke*, the same LLM serves two roles: as a reviewer it scores the quality of the prompt; and as an author it refines the prompt, taking the feedback of the reviewer into account. We further propose a data selection strategy, where we only expose the hard samples to the model. Extensive experiments show that *Evoke* outperforms existing automatic prompt engineering approaches.

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

## A   INSTRUCTION INDUCTION

In *Evoke*, we use the author-reviewer framework to modify a task-specific prompt. In the experiments, we use an off-the-shelf algorithm to generate the initial task-specific prompt. Table 3 demonstrates examples of using instruction induction (Honovich et al., 2022) for prompt initialization.

Table 3: Three examples of instruction inferred from input-output pairs

| Input | Output | Inferred Instruction |
|-------|--------|----------------------|
| Departure | Arrival | Get antonym |
| I am Mike | Ich ben Mike | Translate to German |
| Build | Built | Get passive voice of the given verb |

## B   PROMPTS OF LLM ROLES IN EVOKE

### B.1   LLM-REVIEWER

**Prompt for LLM-Reviewer**

As an experienced teacher, you are well-versed in discerning effective instruction that guides students toward correct answers. Please rate the following instruction on a scale of 1 to 10, where 10 represents the highest level of clarity in problem description, execution steps, and a comprehensive explanation of the problem.
The task at hand is titled: {description}
History that may help you: {memory}
The instruction to be rated is as follows: {instruction}
Kindly provide your rating below.

### B.2   LLM-AUTHOR

**Prompt for LLM-Author**

Task Instruction: {instruction}
We've provided pairs consisting of inputs, the teacher's correct answers, and the students' responses. Please review the incorrect responses from the students and summarize key points that could be adjusted in the instruction to enhance student accuracy.
Pairs: {pairs}
History that may help you: {memory}
To improve the outcome, please revise the task instruction. Highlight major edits and present the updated task instruction.

## B.3 LLM-SELECTOR

**Prompt for LLM-Selector**

As an experienced teacher with insight into the various levels of difficulty of exam questions, please rate the following question on a scale of 1 to 10, considering factors such as conceptual understanding, application of knowledge, problem-solving skills, time required, clarity of language, and accessibility, where 1 denotes extremely easy and 10 denotes extremely difficult.

Task instruction: {instruction}

Input: {input}

Correct answer: {answer}

## C   GENERATED INSTRUCTIONS

We include generated instructions from all tasks below.

### C.1   ORTHOGRAPHY STARTS WITH

**Prompt from Evoke**

Given an input sentence and a specified letter, identify the word or words starting with the given letter. If there are two or more words in a sequence starting with the specified letter, include all of them as a single answer. Ensure to present the word or group of words.
Here are the steps to follow:
-Read the provided input sentence carefully.
-Identify the word or words that start with the specified letter.
-If there are consecutive words starting with the specified letter, group them together as one entity.
-For example, if the input is "I prefer eating apples." and the specified letter is [e], your answer should be eating.

### C.2   COMMON CONCEPT

**Prompt from Evoke**

Given a list, find the commonality between the inputs. The commonality should be a meaningful characteristic, property, or relation that applies to all the inputs, not just a superficial or coincidental feature.
For example, can be used for repairs is a valid commonality for ['sewing', 'wrenches', 'glue', 'surgery'], but tools or skills for joining is too broad and vague, and contain the letter e is too trivial and irrelevant.

### C.3   RHYMES

**Prompt from Evoke**

For this task, you are required to find a word that rhymes with the given word. The word you provide should not be the same as the given word, and should be a real, correctly spelled word from the English language. A rhyming word is defined as a word that has the last syllable sounding identical to the last syllable of the given word. For example, if the given word is "hat", a word that rhymes with it is "cat".
Here are the steps to complete this task:
1. Read the given word carefully.
2. Think of a word that has the same ending sound as the given word.
3. Ensure that the word you thought of is a real word, is spelled correctly, and is not the same as the given word.
4. Write down the rhyming word next to the given word.
Now, please proceed with finding a word that rhymes with each of the following words.

### C.4 MOVIE RECOMMENDATION

---

**Prompt from Evoke**

Given user's interest in movies he watched previously: 'watched'. Now given four different movies from A to D, please recommend one that might be the most interest of the user.
To help you make a good recommendation, consider the following factors:
- The genre, theme, and tone of the movies. For example, if the user likes comedy, action, or drama.
- The similarity or difference between the movies and the ones the user watched before. For example, if the movies are part of a series, a remake, or a spin-off.
- The popularity, ratings, and reviews of the movies. For example, if the movies are critically acclaimed, award-winning, or have a large fan base.
Use these factors to compare and contrast the movies and explain why you think one of them is the best choice for the user. Do not just pick a movie based on your personal preference or guesswork.
Example: If the user watched The Godfather, The Godfather Part II, and Goodfellas, and the options are A) The Departed, B) Scarface, C) The Irishman, and D) Casino, a possible answer is: A
The Departed is a crime thriller that has a similar genre, theme, and tone to the movies the user watched before. It is also a remake of a Hong Kong film called Infernal Affairs, which adds a twist to the familiar story of undercover agents and mobsters. The Departed is a highly popular and acclaimed movie that won four Oscars, including Best Picture and Best Director. It has a star-studded cast that includes Leonardo DiCaprio, Matt Damon, Jack Nicholson, and Mark Wahlberg. The user might enjoy the suspense, the plot twists, and the performances of the actors in this movie. Therefore, I recommend The Departed as the best option for the user.

---

### C.5 LOGICAL FALLACY DETECTION

---

**Prompt from Evoke**

In this task, you are required to identify both informal and formal logical fallacies in the provided input statements. Your response should be a binary value: return 1 if the query is logically valid (i.e., free from any logical fallacies), and return 0 if the query is logically invalid (i.e., contains at least one logical fallacy). A logical fallacy refers to an error in reasoning. Informal fallacies are often content-dependent, such as appealing to irrelevant authority or making hasty generalizations. Formal fallacies, on the other hand, are structural errors in reasoning that occur regardless of the content. It is crucial to consider the structure and the substance of the argument, evaluating whether the conclusions follow logically from the premises, and whether the premises and assumptions are factual and valid. Be cautious not to let personal beliefs interfere with your analysis. For each given pair, compare the input statement against the principles of logical reasoning, to determine whether it contains a logical fallacy or not. Ensure your answer reflects the presence or absence of logical fallacies, thus determining the logical validity or invalidity of the statement. Here are some common examples of logical fallacies:
- Ad Hominem: Attacking the character of a person making an argument rather than the argument itself.
- Appeal to Nature: Claiming something is good because it's natural, or bad because it's unnatural.
- Hasty Generalization: Making a broad claim based on a small or unrepresentative sample size.
- Post Hoc: Assuming that because one event followed another, the first event caused the second event.
- False Cause: Assuming a false or misleading cause-and-effect relationship.

---

## C.6    PRESUPPOSITIONS AS NLI

**Prompt from Evoke**

Determine whether the first sentence entails, contradicts, or is neutral to the second sentence. The term "entailment" means that the information in the first sentence logically supports or leads to the conclusion presented in the second sentence. The term "contradiction" means that the information in the first sentence logically opposes or disproves the information in the second sentence. The term "neutral" implies that the information in the first sentence neither supports nor opposes the information in the second sentence; they are unrelated or the relation between them is ambiguous.

It's important to focus on the factual information provided rather than assumptions or external knowledge. Make sure to carefully read both sentences and analyze their logical relation based only on the given text.

Entailment: The information in the first sentence supports the conclusion in the second sentence.

Contradiction: The information in the first sentence opposes or disproves the information in the second sentence.

Neutral: The information in the first sentence neither supports nor opposes the information in the second sentence, or the relation between them is ambiguous.

For each pair, please provide the correct judgment between entailment, contradiction, and neutral, based only on the provided text. Please avoid assumptions and focus solely on the text provided.

## C.7    WINOWHY

**Prompt from Evoke**

In the given text, you are required to evaluate the reasoning provided concerning the identification of the antecedent of a pronoun in a sentence. The antecedent is the noun that the pronoun is referring to. Carefully examine the reasoning to determine if it accurately identifies the antecedent based solely on the information presented within the sentence itself. Here are the steps you should follow:

Read the sentence and the reasoning provided thoroughly.

-Assess whether the reasoning accurately identifies the antecedent of the pronoun based solely on the provided text. Avoid making assumptions or using external knowledge.

-If the reasoning correctly identifies the antecedent of the pronoun, based on the information given in the sentence.

-If the reasoning fails to accurately identify the antecedent of the pronoun or relies on assumptions or external information.

Remember,

Your evaluation should strictly be based on the information provided in the text.

Your goal is to assess the accuracy of the reasoning in identifying the antecedent of the pronoun.

## C.8 EPISTEMIC REASONING

---

**Prompt from Evoke**

---

In this task, your goal is to determine whether the statement in the "Hypothesis" logically follows from the statement in the "Premise." This is known as entailment. If the "Hypothesis" statement is a logical consequence of the "Premise" statement, then it is an entailment. If it is not, then it is a non-entailment.

-Make sure to carefully consider the relations and assumptions mentioned in both the "Premise" and the "Hypothesis" statements.

-The entailment does not depend on the truth of the statements, but rather whether the logic in the "Hypothesis" follows from the "Premise".

-Pay close attention to the wording and structure of the sentences to analyze whether one entails the other.

Examples:

Entailment

Premise: The sun rises in the east. Hypothesis: The sun rises.

Explanation: The Hypothesis is a simplified version of the Premise and does not introduce any new information or contradictions, hence it's an entailment.

Non-entailment

Premise: Sarah believes that all cats are black.

Hypothesis: All cats are black. Explanation: Even though the Hypothesis is expressed in the Premise, it's tied to Sarah's belief and not presented as a fact, hence it's a non-entailment.

Now, review the provided pairs of statements. Determine if the Hypothesis logically follows from the Premise and respond with either entailment or non-entailment.

---

## C.9  ADVERSARIAL SST2

---
**Prompt from Evoke**

Now given a user query, please do a sentiment analysis where 0 represents negative sentiment and 1 represents positive sentiment. Please note that the input may contain typos. These are not intentional and do not reflect the user's actual sentiment. You should try to correct the typos or infer the intended meaning from the context before assigning a sentiment score. For example, 'to merely badv trather than painfzully awfl' could be corrected to 'to merely bad rather than painfully awful' and scored as 0. Similarly, 'l comelling mobion' could be corrected to 'a compelling motion' and scored as 1. Sentiment analysis.

---

## C.10  ADVERSARIAL QQP

---
**Prompt from Evoke**

You will be given a pair of questions and asked to determine whether they are paraphrases of each other. Paraphrases are questions that have the same meaning or ask about the same information, even if they use different words or structures. Please answer with a binary value of 1 if the questions are paraphrases, or 0 if they are not. Please pay close attention to typos, spelling, grammar, and punctuation before answering, as they may affect the meaning of the questions. If you are not sure whether the questions are paraphrases or not, you can use some strategies to help you decide, such as:

- Compare the keywords and topics of the questions. Do they match or relate to each other?  - Rewrite one question in a different way and see if it still conveys the same message as the other question. - Think about the context and purpose of the questions. Are they asking for the same type of information or response?

For example, the questions *What is the capital of France?* and *Which city is the seat of the French government?* are paraphrases, because they both ask about the same fact and can be answered with the same word (Paris). However, the questions *How do you play the guitar?* and *What are some guitar chords?* are not paraphrases, because they ask for different kinds of information and have different levels of specificity.

---

# D  ADDITIONAL EXPERIMENTS

## D.1  INITIAL PROMPT

The influence of initial prompt quality on the efficacy of *Evoke* was an important aspect of our investigation. To thoroughly examine this, we conducted additional experiments using two alternative initial prompts: a task description and an empty string. As illustrated in Table below, *Evoke*'s performance showed relative insensitivity to the quality of the initial prompt. These results, further underscore the robustness of our approach.

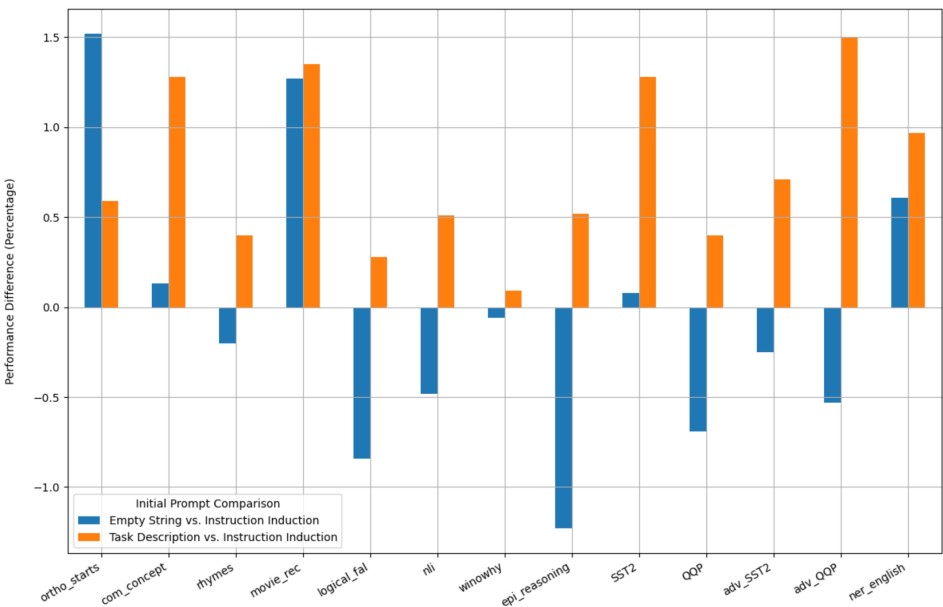

Figure 11: Comparison of different prompt initialization methods: The blue bar indicates that, on average, using a task description as the initialization prompt is more effective, while the orange bar suggests that an empty string performs on par with instruction induction on average.

## D.2 OPEN SOURCE MODELS

We expanded our experiment scope to include open source models Mistral-7B-Openorca and Llama2-70B-Instruct. Our findings reveal notable performance enhancements with these models, affirming the versatility and robustness of our Evoke framework beyond GPT-4.

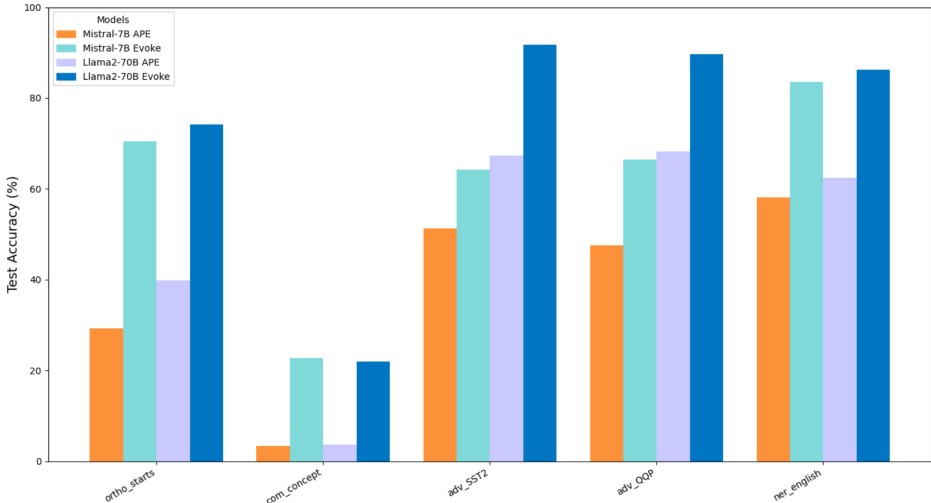

Figure 12: Comparison of test accuracy on APE and Evoke for Mistral-7B-Openorca and Llama2-70B-Instruct: the *Evoke* prompt outperforms the baseline for both open source models.

