# OpenReview forum: "Evoke: Evoking Critical Thinking Abilities in LLMs via Reviewer-Author Prompt Editing"
_ICLR.cc/2024/Conference — ICLR 2024 poster_

### Official Review · Reviewer_QMNd · 2023-10-29

**Soundness:** 2 fair
**Presentation:** 3 good
**Contribution:** 2 fair
**Rating:** 6
**Confidence:** 3

**Summary:**

The paper introduces Evoke, an automatic prompt refinement framework designed to enhance the performance of large language models (LLMs). Traditional prompting methods often underutilize the potential of LLMs due to ad-hoc prompt selection or inefficient random search. Evoke employs an author-reviewer feedback loop with two instances of an LLM: one acting as a reviewer and scoring the current prompt, and the other as an author, which refines the prompt based on feedback and edit history. Additionally, Evoke incorporates a data selection approach that exposes the LLM to more challenging samples, allowing for deeper task understanding.

**Strengths:**

The proposed method outperforms state-of-the-art approaches.

The paper is well-written and easy to follow.

The approach is not human labor intensive.

The self improving adversarial setting is interesting.

**Weaknesses:**

Not enough details are provided about loss functions of each module.

Not enough explanation of the modules; making it difficult to reproduce for other researchers.

**Questions:**

What are the loss functions for each module?

What are the targets for example for first difficulty ranking module?

---

> ### Author Response · Authors · 2023-11-18
> **Response to reviewer QMNd**
>
> We are grateful for the reviewer's detailed observations, which have prompted us to provide clearer explanations. Here are our responses to the concerns raised:
>
> ***Regarding Loss Functions for Each Module:*** `Not enough details are provided about loss functions of each module.`
>
> ***Response***: We apologize for any confusion caused by our initial explanation. The update process for each module in our system is primarily driven by prompting a large language model (LLM), such as GPT-4. The specific prompts used for the different LLM roles within the Evoke framework are detailed in Appendix B. Additionally, the exact implementation process for these updates is clearly outlined in Algorithm 1 of our paper.
>
> ***Explanation and Reproducibility of Modules:*** `Not enough explanation of the modules; making it difficult to reproduce for other researchers.`
>
> ***Response***: We understand the importance of reproducibility in research. To this end, we have provided a comprehensive description in Algorithm 1. Moreover, the prompts used for each module's role are included in Appendix B. We are committed to enhancing the reproducibility of our work and will make the code and data publicly available alongside the publication of the camera-ready version of our paper.
>
> ***Loss Functions for Each Module:*** `What are the loss functions for each module?`
>
> ***Response***: As previously mentioned, our method, Evoke, does not incorporate traditional explicit loss functions due to its reliance on prompting an LLM. However, it's noteworthy that recent studies have begun to explore the connection between in-context learning in LLMs and traditional gradient learning methods [1]. Additionally, there's emerging research that explores understanding chain-of-thought prompting from a gradient perspective [2]. In Evoke, the instruction formulation for LLM-Author could be interpreted as a synergistic integration of in-context learning and chain-of-thought prompting. This suggests the possibility of an implicit, or 'proxy', loss function guiding the update process, aligning with these recent insights into LLM behaviors.
>
>
> ***Targets for the LLM-Selector Module:*** `What are the targets for example for first difficulty ranking module?`
>
> ***Response***: The primary objective of the LLM-Selector module is to assess the difficulty level of a given task, considering the instruction, input, and correct answer. This assessment is then used to identify and select the most challenging ('Hard') examples. We focus on hard examples because LLMs, due to their extensive training on diverse datasets, are typically proficient in solving easier tasks. Incorporating easy examples into our prompt editing process would not significantly enhance its learning process. This selection strategy is thoroughly analyzed in the Experiment section of our paper, where we compare the outcomes of selecting 'Hard,' 'Easy,' and 'Random' examples. We welcome any further inquiries on this aspect of our research.
>
> **References:**
>
> [1] Dai, Damai, Yutao Sun, Li Dong, Yaru Hao, Zhifang Sui, and Furu Wei. "Why can gpt learn in-context? language models secretly perform gradient descent as meta optimizers." arXiv preprint arXiv:2212.10559 (2022).
>
> [2] Wu, Skyler, Eric Meng Shen, Charumathi Badrinath, Jiaqi Ma, and Himabindu Lakkaraju. "Analyzing chain-of-thought prompting in Large language models via gradient-based feature Attributions." arXiv preprint arXiv:2307.13339 (2023).

---

> > ### Author Response · Authors · 2023-11-22
> > **Follow Up**
> >
> > We greatly appreciate the time and effort you've dedicated to reviewing our paper! Should there be any remaining questions or concerns following our response, we would be happy to provide further clarification during the discussion period.

---

### Official Review · Reviewer_7Mrx · 2023-10-29

**Soundness:** 2 fair
**Presentation:** 2 fair
**Contribution:** 3 good
**Rating:** 6
**Confidence:** 3

**Summary:**

This paper describes a method for refining prompts using GPT4. It uses GPT4 in three settings: as an author (to refine prompts) as a Reviewer (to assess quality of prompts and give feedback) as a selector (to select good examples to include in prompts).

The author instance takes as input the current prompt, and the historical feedback to create a new prompt. The reviewer takes the previous prompts, end-task accuracy when using the prompts to assess quality of the new prompt. Lastly, the selector identifies instances that would be difficult for the model to answer given the current prompt and these instances are used by the author and the reviewer.

 Experiments have been presented with qualitative examples from the Instruction Induction datasets. Comparisons against human-generated prompts, and a baseline method for automated prompt engineering ( a method based on paraphrasing existing prompt candidates) reveal significant improvements on multiple tasks including logical fallacy detection, movie recommendation, common concept identification. Experiments on perturbed prompts reveal that the method also results in improved robustness towards typographical errors as it ends up generating instructions that explain how those should be handled by the model. The paper also presents an analysis indicating a correlation between the Reviewer scores and the end-task accuracy. It also shows that as the cycle repeats, the author generates better prompts (as determined improved end-task accuracy) and that the use of the selector for identifying difficult instances help improve performance.

**Strengths:**

- Simple method to improve the performance of prompts
- Experiments on instruction induction tasks show a significant improvement

**Weaknesses:**

- Writing needs to be improved in some parts (more details in questions)
- Studies only GPT4 - unclear if there are any takeaways from the paper if one isn't using GPT4.  (see question)

**Questions:**

1.  The choice of using a reviewer to estimate scores instead of directly relying on end-task scores is an interesting one? Is it for computational reasons/costs to query GPT4? An elaboration would help (I may have misunderstood this as aspect). I had to re-read the sections describing the author, reviewer, selector multiple times along with the algo block to be sure of whats going on. This section could benefit from some more details in an image and/or writing.
2. While the paper only uses GPT4, I was wondering if the authors could experiment with open-access models such as Falcon 180B and other smaller-scale models. I am sure that the authors would agree "LLMs" cannot do whatever the paper describes in general -- if someone does not use the GPT4 OpenAI model, is there a takeaway from this paper? A study of how this behavior changes with scale (of model sizes) could be a useful addition.

---

> ### Author Response · Authors · 2023-11-18
> **Response to reviewer 7Mrx 1/2**
>
> We are grateful for your insightful questions, which highlight important aspects of our study. Let us provide further clarification:
>
> ***Regarding the Use of a Reviewer for Scoring:*** `The choice of using a reviewer to estimate scores instead of directly relying on end-task scores is an interesting one? Is it for computational reasons/costs to query GPT4? An elaboration would help (I may have misunderstood this as aspect). I had to re-read the sections describing the author, reviewer, selector multiple times along with the algo block to be sure of whats going on. This section could benefit from some more details in an image and/or writing.`
>
> ***Response***: We appreciate your observation about our use of the LLM-Reviewer. Our decision was driven by both efficiency and effectiveness:
>
> - **Correlation with End-Task Accuracy:**
>
> One significant challenge in evaluating auto-prompt tasks, such as those in Instruction Induction, Big Bench, and the OpenLLM Leaderboard, lies in the limited size of the datasets. Typically, these datasets contain fewer than 1,000 entries, a stark contrast to the tens or hundreds of thousands of data points commonly used in traditional machine learning. This small sample size inevitably leads to high variance and instability in performance metrics when using a standard train-test split. As a result, the derived accuracy from such a small dataset may not reliably reflect the true task performance.
>
> In contrast, LLMs like the one we employ in LLM-Reviewer are trained on vast and diverse data sources, encompassing a wide range of tasks. This extensive training equips LLMs with a robust understanding and capability to assess task performance more reliably. Our approach leverages this strength of LLMs. As shown in Figure 9, our analysis reveals a significant positive correlation between the evaluations made by LLM-Reviewer and the actual end-task accuracy. This correlation not only validates our methodology but also underscores the reliability of LLM-Reviewer as a proxy for task performance assessment.
>
> Moreover, as evidenced in Table 1 below, the LLM-Reviewer outperforms the Author-Only approach in accuracy and stability. This further emphasizes the advantage of integrating LLM-Reviewer in the evaluation process, particularly when dealing with tasks characterized by limited data availability. By utilizing LLM-Reviewer, we are able to mitigate the challenges posed by small datasets and provide a more accurate and stable measure of task performance.
>
> - **Computational Efficiency:**  Opting for the LLM-Reviewer's score rather than direct task accuracy for prompt selection notably enhances computational efficiency. This method reduces the frequency of querying the inference LLM (e.g., GPT-4), thus saving computational resources.
>
> In the camera-ready version of the paper, we will use the additional one page to expand on these motivations and provide additional details for clearer understanding. We also plan to include a comparative analysis showing the performance enhancement when incorporating the reviewer, as outlined in the table below:
>
> Table 1: Test Accuracy (%) Change on **Author-Reviewer vs. Author-only**
>
> | Task | Author-Reviewer vs. Author-only |
> | -------- | ------- |
> | orthography starts with | +5.8%|
> | common concept | +1.9% |
> | rhymes | +3.1% |
> | movie recommendation | +4.3% |
> | logical fallacy detection | +3.9% |
> | presuppositions as nli | +1.5% |
> | winowhy | +3.6% |
> | epistemic reasoning | +4.2% |
> | SST-2 | +4.7% |
> | QQP | +2.4% |
> | adversarial SST-2 | +4.9% |
> | adversarial QQP | +5.1% |
> | NER (English) | +3.4% |

---

> > ### Author Response · Authors · 2023-11-18
> > **Response to reviewer 7Mrx 2/2**
> >
> > ***On Experimenting with Other LLMs:*** `While the paper only uses GPT4, I was wondering if the authors could experiment with open-access models such as Falcon 180B and other smaller-scale models. I am sure that the authors would agree "LLMs" cannot do whatever the paper describes in general -- if someone does not use the GPT4 OpenAI model, is there a takeaway from this paper? A study of how this behavior changes with scale (of model sizes) could be a useful addition.`
> >
> > ***Response***: Your question about the generalizability of our approach with different LLMs is very pertinent. We agree that not all LLMs exhibit the same level of proficiency in instruction-following. It's also crucial to note that model performance is influenced not only by size but also by factors like training methods and data curation. These aspects significantly impact a model's ability to follow instructions. To address this:
> >
> > - **Additional Experiments with Diverse Models:** To further validate the versatility of our approach, we have extended our experimentation to include models such as Mistral-7B and Llama2-70B. Mistral-7B stands out as a top-performing model in the category of those under 30 billion parameters. Meanwhile, Llama2-70B represents larger-scale models, offering a comprehensive view across a broad spectrum of model sizes. These additional tests are designed to demonstrate the adaptability and robustness of our method, across model scale and inherent capabilities, thereby underscoring its broad applicability. We report the results in Table 2 and 3 below.
> >
> >
> > Table 2: Test Accuracy (%) Comparison on APE and Evoke for **Mistral-7B**
> >
> >
> > | Task | APE | Evoke |
> > | -------- | ------- | ------- |
> > | orthography starts with | 29.3 |70.4|
> > | common concept | 3.3|22.7|
> > | rhymes | 66.9 | 74.7|
> > | movie recommendation | 26.4 | 72.7|
> > | logical fallacy detection | 48.4| 77.3 |
> > | presuppositions as nli | 21.6| 29.1|
> > | winowhy |18.2| 60.9 |
> > | epistemic reasoning | 52.6 | 69.3 |
> > | SST-2 |62.2 |70.0 |
> > | QQP |54.1| 73.1 |
> > | adversarial SST-2 | 51.3 | 64.2 |
> > | adversarial QQP | 47.5 |66.4 |
> > | NER (English) | 58.1 |83.5 |
> >
> > Due to limited time and computational resources, we have not finished all experiments on Llama2-70B, we'd like to firstly present available result below.
> >
> > Table 3: Test Accuracy Comparison (%) on APE and Evoke for **Llama2-70B**
> >
> >
> > | Task | APE | Evoke |
> > | -------- | ------- | ------- |
> > | orthography starts with | 39.8 |74.2|
> > | common concept | 3.6 | 21.9 |
> > | adversarial SST-2 | 67.3 |91.7 |
> > | adversarial QQP | 68.2 |89.6 |
> > | NER (English) |62.5|86.3 |
> >
> > In summary, the applicability of Evoke extends beyond GPT-4, demonstrating compatibility with a wide range of commonly used open-source LLMs.
> >
> > We are grateful for the reviewer's insightful comments, which have provided us with an opportunity to enhance the clarity, detail, and overall effectiveness of our method. We welcome any further questions and are eager to provide additional information as needed.

---

> > > ### Author Response · Authors · 2023-11-22
> > > **Follow Up**
> > >
> > > We greatly appreciate the time and effort you've dedicated to reviewing our paper! Should there be any remaining questions or concerns following our response, we would be happy to provide further clarification during the discussion period.

---

### Official Review · Reviewer_PdAc · 2023-11-01

**Soundness:** 3 good
**Presentation:** 2 fair
**Contribution:** 2 fair
**Rating:** 5
**Confidence:** 4

**Summary:**

This article introduces an innovative approach to optimizing the prompts for large language models. It presents the "Evoke" method, which combines three instances of the language model to generate, verify, and select prompts for each example. After several iterations, the generated prompts guided by the performance on the training set will generalize well to the test set. The experimental results demonstrate that Evoke outperforms several baseline methods on challenging datasets.

**Strengths:**

1. The problem of automatic prompt generation addressed in this paper holds significant importance in the age of Large Language Models (LLMs). While LLMs can generalize across various tasks, the quality of the generated prompts plays a pivotal role in overall task performance. Therefore, I believe that researching this problem is both important and intriguing.

2. The proposed Evoke method offers a straightforward yet highly effective solution. It seamlessly combines different types of LLMs to create a versatile pipeline for task resolution, all without the need for additional training phases while maintaining exceptional performance.

3. The analysis of Evoke's robustness and suitability for fine-grained tasks underscores its versatility and applicability in a wide range of scenarios.

**Weaknesses:**

1. The Evoke method shares significant similarities with "Large Language Models as Optimizers" from DeepMind. While there are variations in the detailed processes, the core contributions and methodologies closely resemble each other. Since the DeepMind paper was released before the ICLR deadline, it is imperative to include a discussion of this relationship in the paper.

2. Some of the detailed tables and figures could be relocated to the appendix to allow ample space for a more comprehensive description of the experimental settings and the Evoke method itself. A more detailed explanation of the Evoke process would enhance the paper's clarity.

3. The rationale behind selecting the Instruction Induction and Big Bench Instruction Induction datasets is not clearly articulated. Some of these tasks may be practical, while others could be considered more akin to toy tasks. Additionally, the use of the LLM as merely GPT-4, a black-box large language model, raises the question of whether the Evoke method is equally effective with other large language models, such as Llama 2 or ChatGPT.

4. Comparing Evoke to other baseline methods may be perceived as somewhat unfair, given that Evoke leverages supervised information from the training dataset. This distinction should be acknowledged when discussing the comparative results.

**Questions:**

1. A key concern revolves around the distinctions between Evoke and "Large Language Models as Optimizers" by DeepMind. Clarifying these differences is pivotal for a comprehensive understanding of both approaches.

2. Could you provide more information regarding the number of steps involved in generating prompts for each task? Additionally, is the prompt length substantial for each task?

3. Efficiency may be a concern when dealing with very large training datasets, as it necessitates running the large language model multiple times on the entire training dataset to obtain evaluation scores. A discussion on potential optimizations or trade-offs for such scenarios would be beneficial.

4. Section 4.4 highlights the effectiveness of the LLM-Selector. However, it remains unclear why, if the generated prompt is superior, there isn't consistently better or at least equivalent performance on easier examples. This phenomenon is briefly mentioned without a deeper exploration or analysis. Further elaboration would enhance the paper's clarity.

---

> ### Author Response · Authors · 2023-11-18
> **Response to reviewer PdAc 1/3**
>
> We are deeply appreciative of the reviewer's insightful comments and would like to clarify some aspects of our work in light of the observations made:
>
> ***On Similarities with DeepMind's Work:*** `The Evoke method shares significant similarities with "Large Language Models as Optimizers" from DeepMind. While there are variations in the detailed processes, the core contributions and methodologies closely resemble each other. Since the DeepMind paper was released before the ICLR deadline [Sep 7], it is imperative to include a discussion of this relationship in the paper.`
>
> ***Response***:
> We thank the reviewer for highlighting the need to differentiate our Evoke method from [1]. While we are working on similar problems, significant distinctions exist:
>
> - ***Methodological Distinction:*** In our approach, the Large Language Model (LLM) assumes a novel and dual functionality, acting both as an author and a reviewer. This is a key differentiator from [1], where the LLM's role is confined to that of an author. The implications of this dual role are profound and multifaceted.
>
>   As an author, the LLM generates content, similar to [1]. However, as a reviewer, it also critically evaluates and refines the generated content. This reviewer aspect adds an additional layer of quality control and perspective, enabling a more nuanced and balanced output.
>
>   To further understand this distinction, it's instructive to examine the meta prompt used in our method. Our meta prompt for LLM-Author guides the LLM in content creation and the meta prompt for LLM-Reviewer frames its approach to reviewing and refining that content. This dual engagement in the creative and evaluative processes is fundamental to our approach and distinguishes it significantly from the model used in [1], leading to potentially more robust and well-rounded results.
>
> - ***Efficiency Focus:*** Evoke emphasizes efficiency, optimizing prompts in no more than five steps. This is a stark contrast to the hundreds of steps in [1].
>
> - ***Generated Instruction:*** The comparative analysis between the Evoke system and the method described in [1] reveals a strategic difference in instruction generation, tailored to the complexity and familiarity of the tasks. For instance, in tasks like logical fallacy detection, where terms such as 'formal' and 'informal logical fallacies' might be unfamiliar to a general audience, Evoke prompt incorporates detailed definitions that can ensure clarity and comprehension. This is in stark contrast to the more generic approach observed in [1], where generated prompts resemble a task description. Conversely, for more universally understood tasks like movie recommendations, Evoke smartly refrains from over-elaborating, recognizing the common knowledge base of its users. This adaptive strategy in instruction crafting contributes to better task performance.
>
> Regarding the timing of the referenced paper [1], we acknowledge its contemporaneous nature, as per the reviewer guideline: "We consider papers contemporaneous if they are published (available in online proceedings) within the last four months". And the suggested paper [1] was published less than one month from ICLR deadline. However, we will ensure to discuss its relationship and the nuances differentiating our work in the revised manuscript.
>
>
>
> ***Improving Evoke's Methodological Clarity:*** `Some of the detailed tables and figures could be relocated to the appendix to allow ample space for a more comprehensive description of the experimental settings and the Evoke method itself. A more detailed explanation of the Evoke process would enhance the paper's clarity.`
>
> ***Response***: We agree with the suggestion to relocate some detailed tables and figures to the appendix. This will create space for a more thorough description of the experimental settings and the Evoke method. In the camera-ready version, we will include additional explanatory text to distinguish our main algorithm from [1], ensuring greater clarity for readers.

---

> ### Author Response · Authors · 2023-11-18
> **Response to reviewer PdAc 2/3**
>
> ***Rationale Behind Dataset Selection and LLM Usage:*** `The rationale behind selecting the Instruction Induction and Big Bench Instruction Induction datasets is not clearly articulated. Some of these tasks may be practical, while others could be considered more akin to toy tasks. Additionally, the use of the LLM as merely GPT-4, a black-box large language model, raises the question of whether the Evoke method is equally effective with other large language models, such as Llama 2 or ChatGPT.`
>
>
> ***Response***:
> - **Dataset Choice:** The selection of the Instruction Induction and Big Bench Instruction Induction datasets was strategic, aligning with current practices in auto-prompt work [1,2,3]. These datasets offer a broad spectrum of task difficulties, reflecting real-world LLM applications. We chose them for their relevance and to facilitate comparisons with existing and future studies in auto-prompting.
> - **LLM Choice:** We selected GPT-4 for its state-of-the-art capabilities, as noted in [4]. However, it can also be extended to smaller scale, open source LLMs. To demonstrate this, we conducted additional experiments with Mistral-7B and Llama2-70B for inference, showcasing significant improvements across various tasks. The detailed results are presented in Table 1 of our response.
>
>
> Table 1: Test Accuracy (%) Comparison on APE and Evoke for **Mistral-7B**
>
>
> | Task | APE | Evoke |
> | -------- | ------- | ------- |
> | orthography starts with | 29.3 |70.4|
> | common concept | 3.3|22.7|
> | rhymes | 66.9 | 74.7|
> | movie recommendation | 26.4 | 72.7|
> | logical fallacy detection | 48.4| 77.3 |
> | presuppositions as nli | 21.6| 29.1|
> | winowhy |18.2| 60.9 |
> | epistemic reasoning | 52.6 | 69.3 |
> | SST-2 |62.2 |70.0 |
> | QQP |54.1| 73.1 |
> | adversarial SST-2 | 51.3 | 64.2 |
> | adversarial QQP | 47.5 |66.4 |
> | NER (English) | 58.1 |83.5 |
>
> Due to limited time and computational resources, we have not finished all experiments on Llama2-70B, we'd like to firstly present available result below.
>
> Table 2: Test Accuracy Comparison (%) on APE and Evoke for **Llama2-70B**
>
>
> | Task | APE | Evoke |
> | -------- | ------- | ------- |
> | orthography starts with | 39.8 |74.2|
> | common concept | 3.6 | 21.9 |
> | adversarial SST-2 | 67.3 |91.7 |
> | adversarial QQP | 68.2 |89.6 |
> | NER (English) |62.5|86.3 |
>
> ***On Comparative Fairness with Baseline Methods:*** `Comparing Evoke to other baseline methods may be perceived as somewhat unfair, given that Evoke leverages supervised information from the training dataset. This distinction should be acknowledged when discussing the comparative results.`
>
> ***Response***: We appreciate the concern regarding the fairness of comparisons. It's important to note that other baseline methods also leverage supervised information from training datasets. For human prompt engineers, viewing sample data is necessary to understand the given task, and during prompt iterations, failed cases of a certain prompt are also taken into account. For APE, it uses instruction induction to initialize the prompt, along with sample data. All approaches assume the availability of a given dataset with input-output pairs; where Evoke differs is in its more effective usage of this set.
>
> ***Clarifying Evoke's Distinction from DeepMind's Method:*** `A key concern revolves around the distinctions between Evoke and "Large Language Models as Optimizers" by DeepMind. Clarifying these differences is pivotal for a comprehensive understanding of both approaches.`
>
> ***Response***: As previously mentioned, our method's efficiency and dual role of the LLM in Evoke are key differentiators from [1]. These aspects are crucial for understanding the unique contributions of our work.

---

> > ### Author Response · Authors · 2023-11-18
> > **Response to reviewer PdAc 3/3**
> >
> > ***Details on Prompt Generation Process:*** `Could you provide more information regarding the number of steps involved in generating prompts for each task? Additionally, is the prompt length substantial for each task?`
> >
> > ***Response***:
> > - **Number of Steps:** In our prompt generation process, each 'step' represents a stage in refining or expanding the prompt to better suit the task at hand. For all tasks, we have standardized the process to involve T=5 steps. This decision was grounded in a thorough empirical analysis. Initially, we experimented with a higher count of T=10 steps. However, we observed that beyond the fifth step, there was no significant increase in accuracy - the benefits started to plateau. This finding indicated that five steps struck an optimal balance, effectively enhancing prompt quality without unnecessary complexity. By maintaining this uniformity across different tasks, we not only ensure the comparability of our results but also optimize the efficiency of the process. This approach ensures that each prompt is effective yet concise, tailored to each task without excess verbosity.
> >
> > - **Prompt Length Variation:** The length of the generated prompts indeed varies depending on the specific requirements of each task. For instance, the logical fallacy detection task necessitates detailed definitions within the prompt, while simpler tasks like movie recommendations do not.
> >
> > ***Efficiency in Large Training Datasets:*** `Efficiency may be a concern when dealing with very large training datasets, as it necessitates running the large language model multiple times on the entire training dataset to obtain evaluation scores. A discussion on potential optimizations or trade-offs for such scenarios would be beneficial.`
> >
> > ***Response***: We acknowledge that efficiency is a concern, especially with large datasets. However, using the generated prompt to obtain its performance on a training set is common for all auto-prompt works (including our baseline [2] and the work recommended by the reviewer [1]), as this process is used to evaluate the prompt during prompt editing.
> >
> > ***On LLM-Selector Effectiveness and Data Difficulty:*** `Section 4.4 highlights the effectiveness of the LLM-Selector. However, it remains unclear why, if the generated prompt is superior, there isn't consistently better or at least equivalent performance on easier examples. This phenomenon is briefly mentioned without a deeper exploration or analysis. Further elaboration would enhance the paper's clarity.`
> >
> > ***Response***: Sorry for the confusion; The difference in performance between 'Hard' and 'Easy' samples is a deliberate part of our methodology to showcase the effectiveness of our selection process.
> >
> > In our approach, we categorize samples into 'Hard' and 'Easy' based on their difficulty level:
> >
> > In our study, 'Hard' samples are chosen for their high level of difficulty, while 'Easy' samples represent those with the lowest difficulty. The LLM-Author utilizes these samples to refine and edit prompts. The difficulty scores for these samples are determined using the LLM-Selector, with the specific prompts for this process detailed in Appendix B.
> >
> > We find that this differentiation between 'Hard' and 'Easy' is crucial in demonstrating the versatility of our model. The significant drop in performance when the selection is reversed (i.e., when 'Hard' methods are applied to 'Easy' samples and vice versa) validates our selection method's effectiveness. This finding underscores the importance of appropriate sample selection in optimizing model performance, a point we discuss in greater detail in Section 4.4.
> >
> > We hope this explanation clarifies the rationale behind our methodological choices and their impact on the model's performance. We remain open to further discussions and insights that can refine our approach.
> >
> > **References:**
> >
> > [1] Yang, Chengrun, Xuezhi Wang, Yifeng Lu, Hanxiao Liu, Quoc V. Le, Denny Zhou, and Xinyun Chen. "Large language models as optimizers." arXiv preprint arXiv:2309.03409 (2023).
> >
> > [2] Zhou, Yongchao, Andrei Ioan Muresanu, Ziwen Han, Keiran Paster, Silviu Pitis, Harris Chan, and Jimmy Ba. "Large language models are human-level prompt engineers." arXiv preprint arXiv:2211.01910 (2022).
> >
> > [3] Sun, Hong, Xue Li, Yinchuan Xu, Youkow Homma, Qi Cao, Min Wu, Jian Jiao, and Denis Charles. "AutoHint: Automatic Prompt Optimization with Hint Generation." arXiv preprint arXiv:2307.07415 (2023).
> >
> > [4] https://huggingface.co/spaces/lmsys/chatbot-arena-leaderboard

---

> > > ### Author Response · Authors · 2023-11-22
> > > **Follow Up**
> > >
> > > We greatly appreciate the time and effort you've dedicated to reviewing our paper! Should there be any remaining questions or concerns following our response, we would be happy to provide further clarification during the discussion period.

---

### Official Review · Reviewer_hcVA · 2023-11-03

**Soundness:** 4 excellent
**Presentation:** 4 excellent
**Contribution:** 4 excellent
**Rating:** 8
**Confidence:** 4

**Summary:**

This work proposed an author-reviewer framework, Evoke, for automatic prompt engineering. In this framework, the same LLM serves two roles: as a reviewer it scores the quality of the prompt; and as an author it refines the prompt, taking the feedback of the reviewer into account. On top of this, the authors further propose a data selection strategy, where they only expose the hard samples to the model. This work also conduct extensive experiments to demonstrate that Evoke outperforms existing automatic prompt engineering approaches.

====After authors' discussion===
I have read through the authors' response, and I think they have addressed all my concerns. This is a nice work, and many thanks for the efforts!

**Strengths:**

[+] The Evoke framework is novel in its approach to improving LLM prompt refinement using an author-reviewer framework (with a data selector).

[+] The performance improvements seems impressive especially on some very difficult tasks.

**Weaknesses:**

[-] Iteration-Dependent: The efficacy of the approach may heavily depend on the number of iterations and the quality of feedback loops, which could vary significantly based on the complexity of the task and the initial prompt quality.

[-] Complexity and Overhead: The Evoke model might introduce additional complexity and computational overhead since we need to do the iterative selection process T times.

**Questions:**

- Are all the prompts generated by Evoke concatenated together during the evaluation phase?

- How do the quality of initial prompts affect the effectiveness of the Evoke method throughout iterations?

---

> ### Author Response · Authors · 2023-11-18
> **Response to reviewer hcVA**
>
> **We sincerely thank the reviewer for recognizing the novelty and effectiveness of our method. We appreciate the opportunity to clarify and expand upon the questions raised:**
>
> ***Concerning Iteration Dependence:*** `The efficacy of the approach may heavily depend on the number of iterations and the quality of feedback loops, which could vary significantly based on the complexity of the task and the initial prompt quality.`
>
> ***Response***: We appreciate your concern about the iteration-dependent nature of our approach. To clarify, we have standardized the number of iterations (T) to 5 across all tasks in our experiments. We have initially tried T up to 10, and setting T=5 was informed by preliminary experiments showing converging improvements in task accuracy at this iteration count, irrespective of task complexity. Furthermore, we experimented with varying initial prompts, including task descriptions and empty strings, to assess the sensitivity of our Evoke algorithm to initial prompt quality. Our findings, detailed in Table 1 of our response, demonstrate that Evoke's performance is robust to variations in initial prompt quality, consistently outperforming baseline methods.
>
> ***On Complexity and Overhead:*** `The Evoke model might introduce additional complexity and computational overhead since we need to do the iterative selection process T times.`
>
> ***Response***: We acknowledge the reviewer's point about potential complexity and computational overhead. To contextualize this, we compared Evoke's process with the conventional human prompt engineering workflow, which typically involves several iterative steps of testing, analyzing, and refining prompts. Interestingly, Evoke operates more efficiently in terms of computation time while achieving comparable or superior accuracy. This efficiency gain is particularly notable given that Evoke's iterative process, even set at its most computationally intensive mode, matches the computational demands of a single iteration in the human-driven process. Furthermore, by limiting T to 5 in our experiments, we achieved a balanced trade-off between performance optimization and computational resource usage.
>
> Additionally, in all experiments, we set T to 5, resulting a strong optimized prompt.
>
> ***Regarding Evaluation Phase and Prompt Concatenation:*** `Are all the prompts generated by Evoke concatenated together during the evaluation phase?`
>
> ***Response***:
>
> - **During Training (Prompt Editing) Evaluation Phase:** In the training phase of our method, each prompt generated by Evoke undergoes an independent evaluation process. We have adopted this strategy deliberately to mitigate any potential interactions or biases that could arise from prompt concatenation. This independent assessment of each prompt allows us to accurately gauge the effectiveness of every individual prompt generated, ensuring that our method's learning process is informed by clear, unbiased feedback.
>
> - **During Inference Evaluation Phase:** For the evaluation phase during inference, our approach is different. Here, we utilize only the final optimized prompt generated by Evoke for calculating accuracy on the test set. This decision is based on our objective to evaluate the practical applicability of the final prompt in real-world scenarios. By focusing on the final prompt, we are able to assess the ultimate effectiveness of the Evoke algorithm in producing a single, highly effective prompt for practical use.
>
> ***Impact of Initial Prompt Quality:*** `How do the quality of initial prompts affect the effectiveness of the Evoke method throughout iterations?`
>
> ***Response***: The influence of initial prompt quality on the efficacy of Evoke was an important aspect of our investigation. To thoroughly examine this, we conducted additional experiments using two alternative initial prompts: a task description and an empty string. As illustrated in Table 1, Evoke's performance showed relative insensitivity to the quality of the initial prompt. These results, further underscore the robustness of our approach.
>
> Table 1: Change in Task Accuracy Using Different Initial Prompts
>
> | task | empty string vs. instruction induction | task description vs. instruction induction |
> | -------- | ------- | ------- |
> | orthography starts with | 1.52%| 0.59%|
> | common concept | 0.13% | 1.28% |
> | rhymes | -0.20% | 0.40% |
> | movie recommendation | 1.27% | 1.35% |
> | logical fallacy detection | -0.84% | 0.28% |
> | presuppositions as nli | -0.48% | 0.51% |
> | winowhy | -0.06% | 0.09% |
> | epistemic reasoning | -1.23% | 0.52% |
> | SST-2 | 0.08% | 1.28% |
> | QQP | -0.69% | 0.40% |
> | adversarial SST-2 | -0.25% | 0.71% |
> | adversarial QQP | -0.53% | 1.50% |
> | NER (English) | 0.61% | 0.97% |
>
> We hope these responses address the reviewer's concerns adequately and demonstrate the robustness and efficiency of our Evoke method. We remain open to further discussions and insights that can refine our approach.

---

> > ### Author Response · Authors · 2023-11-22
> > **Follow Up**
> >
> > We greatly appreciate the time and effort you've dedicated to reviewing our paper! Should there be any remaining questions or concerns following our response, we would be happy to provide further clarification during the discussion period.

---

### Author Response · Authors · 2023-11-18
**General Response**

We sincerely appreciate the time and effort invested by all reviewers in evaluating our paper. Your insightful feedback has been instrumental in refining our work. Here, we list the modifications implemented in response to the collective feedback, alongside specific responses to each reviewer's comments.

- **[Experiments] Applicable to Open Source Models (e.g., Mistral-7B and Llama2-70B):**

Addressing the curiosity of reviewers PdAc and 7Mrx, we expanded our experiment scope to include open-source models [Mistral-7B](https://huggingface.co/Open-Orca/Mistral-7B-OpenOrca) and [Llama2-70B](https://huggingface.co/upstage/Llama-2-70b-instruct). Our findings reveal notable performance enhancements with these models, affirming the versatility and robustness of our Evoke framework beyond GPT-4.

- **[Experiments] Diverse Initial Prompt Strategies:**

Responding to reviewer hcVA's point on initial prompt quality, we incorporated experiments with task descriptions and empty strings as initial prompts. This approach underscores the feasibility of Evoke with accessible, low-effort prompt initiation methods.


- **[Analysis] Author-Reviewer vs. Author-Only Dynamics:**

In line with the Evoke framework, where authors edit prompts and reviewers evaluate them, we conducted a comparative analysis between author-reviewer collaboration and author-only setups. This analysis highlights the critical role of the reviewer in enhancing the efficacy of Evoke.

- **[Clarifications] Enhanced Experimental Details:**
  - **Algorithm and Workflow:** We have clarified the implementation of the Evoke workflow (outlined in Algorithm 1) and included details about the prompt for each role in Appendix B, facilitating reproducibility.
  - **Evoke Effectiveness Across Model Sizes:** We now present comprehensive results demonstrating Evoke's effectiveness varying with model sizes, in our response to reviewer PdAc and 7Mrx.

- **[Clarification] Methodology Elaboration:**
  - We have provided an in-depth explanation of each role within the Evoke framework, thereby clarifying our methodology and its implications, in our response to reviewer 7Mrx and QMNd.


We appreciate the reviewer's insightful feedback, as it has given us a chance to improve the clarity, detail, and overall effectiveness of our work. We are open to any further questions and ready to provide additional information if needed.

---

### Author Response · Authors · 2023-11-21
**Follow Up**

We greatly appreciate the time and effort you've dedicated to reviewing our paper! Should there be any remaining questions or concerns following our response, we would be happy to provide further clarification during the discussion period.

---

### Meta-Review · Area_Chair_RiEU · 2023-12-21

**Metareview:**

This paper proposes an automatic prompt refinement framework called Evoke, which leverages two LLMs, one as a reviewer and the other as an author. The LLM-author edits the prompt based on the reviewer's feedback and the edit history, while the LLM-reviewer scores the current prompt. Experimental results show that Evoke can outperform existing methods substantially. Reviewers generally agree that the problem under study is interesting and important, the proposed method is very effective, and the paper is well-written overall. They also pointed out several weaknesses such as the distinction between Evoke and a piece of existing work from DeepMind, complexity and overhead, as well as only using GPT-4 as the LLM. The authors' responses to all concerns and questions seem satisfactory. I strongly suggest that the authors add the new experimental results and revise the paper as promised in the discussion period, should the paper gets accepted.

**Justification For Why Not Higher Score:**

The paper presents an effective framework but it is also straightforward, which makes it not exciting enough to be an oral or spotlight paper. Please also see the weaknesses as mentioned above.

**Justification For Why Not Lower Score:**

Please see the strengths of the paper summarized above. The paper overall makes valuable contributions to automatically optimizing prompts. I am leaning towards accepting this paper, but would not mind if this paper has to get rejected because of other more exciting papers with higher novelty.

---

### Decision · Program_Chairs · 2024-01-16

Accept (poster)